Utilization and transformation of Chrysotila dentata-derived dissolved organic matter by phycosphere bacteria Marinobacter hydrocarbonoclasticus and Bacillus firmus

Wang Xueru 1 2 3
Fan Chenjuan 3
Sun Jun phytoplankton@163.com 1 2 3
1 China University of Geosciences, Institute for Advance Marine Research , Guangzhou , China
2 China University of Geosciences, State Key Laboratory of Biogeology and Environmental Geology , Wuhan , China
3 Tianjin University of Science and Technology, Research Centre for Indian Ocean Ecosystem , Tianjin , China
Brazelton William
Electronic publication date: 2024 Jan 4
Publication date: 2024
Volume: 12
Electronic Location ID: e16552
Received 2023 Jun 1; Accepted 2023 Nov 9
Copyright: ©2024 Wang et al.
Copyright year: 2024
Copyright holder: Wang et al.
License: This is an open access article distributed under the terms of the Creative Commons Attribution License, which permits unrestricted use, distribution, reproduction and adaptation in any medium and for any purpose provided that it is properly attributed. For attribution, the original author(s), title, publication source (PeerJ) and either DOI or URL of the article must be cited.
License URL: https://creativecommons.org/licenses/by/4.0/

Keywords: Chrysotila dentata-derived DOM, Marinobacter hydrocarbonoclasticus, Bacillus firmus, EEM-PARAFAC, Fluorophores, Spectroscopic indices

Funding: National Key Research and Development Project of China 2019YFC1407805 Changjiang Scholar Program of Chinese Ministry of Education T2014253 State Key Laboratory of Biogeology and Environmental Geology, China University of Geosciences GKZ21Y645 GKZ22Y656 This research was financially supported by the National Key Research and Development Project of China (2019YFC1407805), the National Natural Science Foundation of China (41876134), the Changjiang Scholar Program of Chinese Ministry of Education (T2014253), and supported by State Key Laboratory of Biogeology and Environmental Geology, China University of Geosciences (No. GKZ21Y645 and GKZ22Y656) to Jun Sun. The funders had no role in study design, data collection and analysis, decision to publish, or preparation of the manuscript.

==============================
The dissolved organic matter (DOM) released from the cocoolithophores (Chrysotila dentata) was studied in laboratory experiments after co-culturing C. dentata with bacteria. Marinobacter hydrocarbonoclasticus (CA6)-γ-Proteobacteria and Bacillus firmus (CF2) were used to investigate the utilization and processing of the DOM derived from C. dentata, utilizing fluorescence excitation-emission matrix (EEM) combined with parallel factor analysis (EEM-PARAFAC), while measuring algal abundance and photosynthetic parameters. The experimental groups consisted of axenic C. dentata groups, filter cultured with bacteria (CA6 or CF2) groups, C. dentata co-cultured with bacteria (CA6 or CF2) groups and axenic bacteria (CA6 or CF2) groups. We then evaluated the processing of DOM by determining four fluorescence indices. The number of C. dentata cells and the photosynthetic capacity of microalgae were enhanced by CA6 and CF2. The main known fluorophores, including humic-like components and protein-like components, were present in all sample. The protein-like component of algal-bacterial co-cultures was effectively utilized by CA6 and CF2. The humic-like components increased at the end of the culture time for all cultures. Meanwhile, the average fluorescence intensity of protein-like in CA6 co-culture with algae was lower than that in CF2 co-culture with algae over time. On the other hand, the average fluorescence intensity of humic-like in CA6 was higher than CF2. However, the total change in fluorescence in humic-like and protein-like of axenic CF2 cultures was lower than that of CA6. Hence, the ability of CA6 to transform microalgal-derived DOM was superior to that of CF2, and CF2’s ability to consume bacterial-derived DOM was superior to that of CA6.

Introduction

Marine dissolved organic matter (DOM) is one of the largest reservoirs of organic carbon in the global carbon cycle and consists of a mixture of molecules with different molecular weights (Hansell, Carlson & Schlitzer, 2012). The roles of photosynthesis by phytoplankton, metabolic activity by heterotrophic bacteria, and photochemistry in DOM transformation within the global carbon cycle have been well established (Maranón et al., 2004; Mopper et al., 1991). Chromophoric dissolved organic matter (CDOM), which is a typical active and colored component of DOM, plays an essential role in controlling luminous flux as it can absorb UV and visible radiation in various bodies of water (Uusikivi et al., 2010; Zhang et al., 2010). Currently, two major types of CDOM can be identified: protein-like compounds (amino acid-like compounds) and humic-like compounds (Coble, 2007). Generally, the fluorescence of protein-like compounds in marine environments is primarily attributed to tryptophan-like signal and tyrosine-like signals, which are generated by the activity of phytoplankton and bacteria (Nieto-Cid et al., 2005; Stedmon et al., 2007; Cammack et al., 2004; Yamashita & Tanoue, 2003). The production of humic-like substances is associated with microbial oxidation and degradation of organic matter as well as exposure UV and blue light (Stedmon & Markager, 2005).

Coccolithophores, which are a group of calcified protists capable of producing an intricate exoskeleton composed of calcium carbonate scales, serve as primary marine producers and contribute to phytoplankton communities in the ocean (Balch, 2018; Peled-Zehavi & Gal, 2021). They account for approximately 15% of marine primary productivity (Liu et al., 2019a). Chrysotila dentata is a species of calcifying coccolithophores found in the Bohai Sea, China (Liu et al., 2019a). The growth of C. dentata involves a calcification process and the ballasting effect of calcite coccoliths, both having significant influence on the biogeochemical cycle of marine carbon (Sun & Jin, 2011). Furthermore, most phytoplankton dependent on interactions with bacteria for essential micronutrients in various aquatic environments (Falkowski et al., 2004; Croft et al., 2005), such as, marine ecosystems (Liu et al., 2019a; Liu et al., 2019b), lakes (Ren et al., 2023), reservoirs (Zhao et al., 2019), etc. Aquatic heterotrophic bacteria have the ability to utilize and transform 10–50% of the organic matter fixed by phytoplankton, which occurs in microscale interactions within the vicinity of individual phytoplankton cells (Seymour et al., 2017). Additionally, Segev et al. (2016) demonstrate that marine bacteria (Phaeobacter inhibin) can grow in co-culture with algal cells, thereby enhancing the growth of the algae.

How do bacteria process phytoplankton-derived dissolved organic matter (DOM)? Previous research has found that microbial cells can utilize individual molecules of labile DOM (LDOM), or the molecular structure of DOM can be modified by extracellular enzyme from bacteria. Additionally, cells can transform organic molecule as part of refractory DOM (RDOM), which is controlled by bacterial processes and consortia and may be affected by Photodegradation. Subsequently, RDOM is preserved as a carbon component in the ocean (Vähätalo & Wetzel, 2004; Ogawa et al., 2001; Hansell, Carlson & Schlitzer, 2012). Furthermore, some researchers have discovered that α-proteobacteria dominate amino acid uptake in the surface ocean, while γ-proteobacteria and Bacteroidetes also utilize amino acids at higher concentrations (Alonso-Sáez et al., 2007). Degradation of DOM can lead to the formation of inorganic substrates (Francko & Heath, 1982). In summary, microbe-algal interactions play a crucial role in promoting global biogeochemical cycles involving carbon, nitrogen, and phosphorus. To investigate the transformation of C. dentata-derived DOM by significant bacteria in the Bohai Sea waters of China, we conducted laboratory experiments using CA6 and CF2. The aim of this experiment was to address this accurately.

In this study, we utilized excitation-emission matrix (EEM) fluorescence spectroscopy and the Parallel Factor Analysis model (PARAFAC) to characterize variations in C. dentata-derived DOM (Murphy et al., 2011). Additionally, we employed several parameters including the biological index (BIX), the Fluorescence index (FI), the humification index (HIX), β/α, Fv/Fm, the bacterial abundance, and C. dentata to further characterize how C. dentata-derived DOM is processed by CA6 and CF2. The experiment examined how two different bacteria utilize C. dentata-derived DOM over a long-term. Then, we established different treatment groups to investigate the effects of CA6 and CF2 on the photosynthetic activity of algal cells and their utilization of dissolved organic matter in solutions. 

Materials and Methods

Chrysotila dentata cultures

Chrysotila dentata was isolated from the Bohai Sea in China and incubated at 25 °C with a photon flux density of 100 µmol m−2 s−1 and a light: dark cycle of 14:10 h in sterile f/2 medium (Lananan et al., 2013). Prior to the experiment, conical flasks were soaked in hydrochloric acid solution for 48 h and washed with Milli-Q water to minimize organic matter input. The conical flasks were than sterilized in an autoclave at 121 °C for 20 min before C. dentata was cultured using artificial seawater (ASW) medium (Lechtenfeld et al., 2015). Meanwhile, after gently shaking each bottle for a minute, 100 µL of C. dentata solution was counted using a hemocytometer under a microscope (Olympus BX51, Tokyo, Japan). At least 200 cells per sample were counted.

Culture of bacterial strains and growth conditions

CA6 and CF2 were isolated from the C. dentata solution using a 3.0 µm PC membrane (47 mm, Millipore, MA, USA). The 16S rRNA gene sequences of CA6 and CF2 were deposited in GenBank (SRR17041861 and SRR170418856), respectively. Additionally, due to their different bacterial lifestyles, CF2 exhibited a free-living (FL) life while CA6 was in particle-attached (PA) mode. To revitalize the bacterial cultures, the bacterial stocks were inoculated onto 2216E agar plates and cultured at 28 °C for 72 h (Green et al., 2004). Subsequently, a monoclonal bacterium was transferred into a liquid medium of 2216E at 28 °C for 72 h with shaking at 150 rpm. Three passages of bacteria cultured through the 2216E liquid medium were carried out before inoculating the experimental cultures.

Absorption spectroscopy

To assess the variation of chromophoric DOM, liquid samples of CDOM were filtered through 0.22 µm polycarbonate membrane filters (47 mm; Millipore, Burlington, MA, USA). Three-dimensional fluorescence excitation-emission matrices (EEM) of CDOM samples were obtained using a fluorescence spectrophotometer (Hitachi F-700, Tokyo, Japan), with the voltage of the photomultiplier tube setting at 700 V. The emission scans (Em) were set from 250 to 500 nm, and the excitation wavelengths (Ex) ranged from 200 to 450 nm. Both Ex and Em slits were set to 5 nm, and the scanning speed was set at 12,000 nm min−1. Each DOM spectrum underwent blank subtraction using Milli-Q ultrapure water in order to remove most of the Raman scatter. After adjustment as previously mentioned (Stedmon & Bro, 2008; Lawaetz & Stedmon, 2009), Raman calibration was performed. PARAFAC modeling was performed on 30 EEMs data using the Openflour package. Two outliers were then removed from the dataset. The non-negativity constraint was applied in this mode. A core consistency test was conducted to validate the model and determine the number of individual fluorescent components. Components 1 and 2 were effectively identified by PARAFAC model. The fluorescence intensity Raman unit (RU) was used to describe all data in Table 1. Four fluorescence-derived indices (biological index (BIX) (Huguet et al., 2009), fluorescence index (FI), a ratio of fluorescing components (β/α) (Parlanti et al., 2000), and humification index (HIX) (Ohno, 2002)) were utilized to characterize DOM properties and compositions. These indices were calculated based on fluorescence analysis as follows:

Table 1 Central regions of EEM fluorescence attributed to different sources compared with previous studies.

Fluorescent components	Ex/Em	Description	Comparison with previous studies	
C1	275/330-340	Protein-like	Amino acid tryptophan (Eder et al., 2022; Schafer et al., 2021)	
C2	250/410	Humic-like	Humic-like: C2 ((255 (355)/410 nm) (Dainard & Guéguen, 2013; Amaral, Romera-Castillo & Forja, 2021)	

(1) BIX=FEx=310nm,Em=380nmFEx=310nm,Em=430nm

(2) FI=FEx=370nm,Em=470nmFEx=370nm,Em=520nm

(3) HIX=FEx=254nm,Em=435−480nmFEx=254nm,Em=300−345nm

(4) βα=FEx=310nm,Em=380nmFEx=310nm,Em=420−435nm.

Experimental setup

Chrysotila dentata cells were cultivated in ASW medium at 25 °C. After 20 days of microalgae cultivation, the concentration of microalgal was approximately (2.68 ± 0.02) × 107 cells L−1. The C. dentata liquid was then centrifuged at 8,000 rpm for 5 min and filtered through 0.22 µm polycarbonate membranes (Millipore, Burlington, MA, USA). The resulting filtrate was considered as the C. dentata-derived DOM fraction. Monoclonal bacterial colonies CA6 and CF2 were cultured in the volume of 40 ml ASW medium at 28 °C for 3 days before adding a volume of bacterial liquid(3 ml) to each experimental treatment (axenic microalgae liquid, C. dentata-derived fraction, and ASW medium). This day of bacterial addition was labeled as day 0 to denote the beginning point of this experiment. All bottles were covered with tin foil, and all experimental groups were incubated in a diurnal incubator with a light-dark cycle of 12:12 h at 25 °C. To isolate the CDOM fraction, liquid from C. dentata was passed through sterilized 0.22 µm polycarbonate membranes (Millipore, Burlington, MA, USA). In order to evaluate the utilization and transformation ability of CF2 and CA6 derived from C. dentata to C. dentata-derived DOM and bacteria-derived DOM, seven different experimental groups were proposed based on the substrate: (1) axenic C. dentata group, (2) C. dentata-CA6 co-cultured group, (3) CA6 cultured with filtrate group; (4) CA6 cultured with artificial seawater group; (5) C. dentata-CF2 co-cultured group, (6) CF2 cultured with filtrate group, and (7) CF2 cultured with artificial seawater group. Each experimental group consisted of three parallel experiments.

Statistical analysis

This study utilized the General Linear Model in SPSS (version 26.0) to examine changes in various sample characteristics and investigate the relationship between variables across seven groups/samples. The amount of change was calculated using Microsoft Excel 2021 (Microsoft, Bellingham, WA, USA). PCA analysis was conducted in MATLAB (version R2016a) to assess variations in C. dentata cell abundance, fluorescence index, and fluorescence intensity. The measured values were expressed as mean ± standard deviation (SD).

Results

Biomass of C. dentata, and chlorophyll fluorescence

In the co-culture state of C. dentata and bacteria, the cell abundance and photosynthetic parameters (Fv/Fm and QY) of C. dentata exhibit different trends (Figs. 1A–1C). After culturing C. dentata in ASW medium for 20 days, CA6 and CF2 were added to the culture on day 20, which was then designated as day 0. Then, after co-culturing C. dentata with CA6 and CF2 for 22 days, respectively. The abundance of microalgal cells in co-cultured groups of C. dentata-CA6 and C. dentat- CF2 showed high variability, with the algal concentration in C. dentata-CA6 co-cultured group increasing from (2.381 ± 1.062) × 104 cells mL−1 to (16.045 ± 1.359) × 104 cells mL−1during culture, while the abundance of algal cells in C. dentata-CF2 co-cultured group initially increased from (5.379 ± 1.633) × 104 cells mL−1 to (8.57 ± 1.753) × 104 cells mL−1 on the first day, then decreased to (2.820 ± 0.860) × 104 cells mL−1 on third day before slowly increasing again to (8.406 ± 1.079) × 104 cells mL−1. However, the axenic C. dentata group decreased from (3.264 ± 0.279) × 104 cells/mL to (3.090 ± 0.230) × 104 cells mL−1. The significant increase in C. dentata abundance in experimental group was caused by co-culturing C. dentata with bacteria. Additionally, a correlation test was conducted using the general linear model to assess the relationship between the number of C. dentata cells and the photosynthesis activity at different stages: the early stage (1–3 days), middle stage (9–11 days), and late stage (17–22 days) (Figs. 1D–1F). The algae-CA6 co-cultured group showed significantly higher numbers compared to the axenic C. dentata group (p < 0.05) in the early stage of culture, and the C. dentata-CF2 co-culture group exhibited significantly higher number than the axenic microalgae group (p <0.001) in the late stage. During the late phase, CA6 demonstrated a greater capability to increase the number of C. dentata cells compared to CF2 (p <0.001) (Fig. 1D).

Figure 1 Variation in cultivation after 22 days, including the abundance of C. dentata growth (A); chlorophyll fluorescence characteristics (B, C); the abundance of microalgae cells from different groups during different periods (D); and changes in photosynthetic parameters of microalgae during different periods (E, F); Each value represents the mean ± SD (n = 3).

Different symbols (e.g., ** and ***) indicate significant differences between co-culture groups of C. dentata and bacteria at p < 0.05 and p < 0.001 levels, respectively.

One of the essential metabolic processes of phytoplankton is photosynthesis. Fv/Fm and QY were utilized to monitor the photosynthesis activity of C. dentata (Figs. 1B and 1C) (Dabrowski et al., 2021). The co-culture groups of C. dentata-CA6 and C. dentata-CF2 showed a gradual increase in Fv/Fm and QY during the early cultivation stage, followed by a decline eleven days later. In contrast, the axenic C. dentata group exhibited a slower decrease in photosynthetic capacity compared to the other groups. In this study, we observed inconsistent change curves for photosynthetic capacity parameters and the cell abundance index of C. dentata. When C. dentata-bacteria were co-cultured in bottles, the photosynthetic capacity of C. dentata exhibited a faster reaction compared to the growth rate of cells (Figs. 1D–1F).

Fluorescence components by PARAFAC

The artificial seawater was used to incubate all experimental treatment groups, in which CDOM components were composed of the release by three microorganisms: C. dentata, CA6, and CF2. The fluorescence patterns of the seven different experimental groups were similar (Fig. 2) and they were compared with earlier studies from the OpenFluor database (Table 1). PARAFAC Component 1 had excitation and emission wavelengths at Ex/Em = 220 (275)/335 nm and was associated with biologically produced protein-like compounds. PARAFAC Component 2 had excitation and emission wavelengths at Ex/Em = 245 (320) /415 nm; it could be categorized as humic-like fluorophores (Osburn et al., 2016).

Figure 2 EEM-PARAFAC plotted the fluorescence components.

C1, protein-like component (A); C2, humic-like component (C); (B) and (D) are corresponding load diagrams; Raman units (RU) are used to characterize the fluorescence intensity.

Changes in fluorescence intensity of the individual component

To further investigate the variation in CDOM concentration among seven groups, a general linear model was used to perform an analysis of variance (ANOVA) on the intensity of DOM fluorescence (Fig. 3). This is because the fluorescence intensity of DOM is directly proportional to its increased concentration (Tian et al., 2020). During incubation, the average concentration of C1 component in the C. detnata-CF2 co-cultured group was significantly higher than that in the axenic algae group, indicating that CF2 facilitated the release of C1 component from the algal cell. No significant difference existed between the C. dentata-CA6 co-cultured group and the axenic algae group, indicating that CA6 could not be conductive to the release of the C1 component from the C. dentata co-cultured with the CA6 group. On the other hand, the average concentration of the C1 component in the bacteria cultured with filtrate group was significantly higher (p < 0.001) than that in the other two groups, indicating that the bacteria (CA6 and CF2) were unable to efficiently process or utilize the C1 component of the filtrate group (Fig. 3B). During the incubation phase, the changes in the total concentration of the C1 component for the seven groups were 0.09497, 0.06221, 0.00171, −0.00341, 0.02520, −0.01427, and −0.10149, respectively (Fig. 3B). The change in total C1 component concentration of the axenic algae group was higher than that of the other experimental groups, indicating that the addition of bacteria effectively consumed the protein-like component.

Figure 3 Changes in fluorescence components.

Average concentration of C1 component (A), The average rate of change C1 and total C1 changes (B), Average concentration of C2 component (C). The average rate of change C2 and total C2 changes (D). Each value represents the mean ± SD (n = 3). Different symbols (e.g., ** and ***) indicate significant differences in the co-culture group of C. dentata and bacteria (p < 0.05, p < 0.001), respectively.

On the one hand, during incubation, there was a significantly higher mean concentration of the C2 component in CA6 when it was co-cultured with the C. dentata compared to the axenic C. dentata group (p < 0.05) (Fig. 3C). This indicates that CA6 facilitated the release of humic-like component from the CA6-C. dentata co-culture system. On the other hand, no significant difference was observed in the mean concentration of the C2 component between CF2 co-cultured with the C. dentata group and the axenic C. dentata group in terms of average, suggesting that CF2 could not facilitate the release of humic-like component from the CF2-C. dentata co-culture system. The CA6 cultured with filtrate group showed a significantly higher level compared to the CA6-C. dentata co-cultured group and the CA6 cultured with ASW group, indicating that CA6 can promote the generation of humic-like components in the filtrate more effectively. The changes in total concentration of the C2 component in the seven groups were as follows: 0.024932, 0.053197, 0.088968, 0.044121, 0.028149, 0.022104, and −0.0032, respectively (Fig. 3D). We observed that CF2 is capable of utilizing bacteria-derived humic-like components. The rate of change in concentration of the C2 component was compared among the seven groups during incubation (Fig. 3D). Except for the C. dentata-bacteria co-cultured group, it was found that the rate of change in C2 concentration was higher in the CA6 treatment groups than in the CF2 treatment groups, suggesting that CA6 has a higher rate of generating humic-like components compared to CF2.

Fluorescence indices

The changes in fluorescence characteristics among seven groups, as indicated by the fluorescence indices (FI, BIX, β/α, HIX), are shown in Fig. 4. Overall, the HIX values ranged from 0.215 to 0.351, the β/α values ranged from 0.799 to 1.396, the FI values ranged from 2.587 to 3.551, and the BIX values ranged from 1.226 to 1.601 across the seven experimental groups. Subsequently, a general linear model was used to test these four fluorescence indices of the experimental groups. It was observed that the HIX index of the C. dentata-CA6 co-cultured group exhibited significantly higher levels compared to those of the axenic algae group (p < 0.05). Amongst the CA6 cultures in experimental groups, both the C. dentata-CA6 co-cultured group and the CA6 cultured with filtrate group had higher HIX indices than that of the axenic CA6 group. On the other hand, the HIX value of the C. dentata-CF2 co-cultured group was significantly higher than that of both CF2 cultured with filtrate and CF2 cultured with ASW groups (p < 0.05). A higher HIX value indicates a greater presence of higher-molecular-weight aromatic compounds, suggesting a more favorable enrichment of humic-like substances in the microalgae-bacteria (CF2 and CA6) co-cultured environment. FI > 1.8 signifies that microorganisms were primarily responsible for producing humic-like components in DOM production, while the β/α index is used to characterize proportions of DOM production and decomposition. BIX can serve as an indicator of DOM traceability in water, with its values also indicating the presence of DOM derived from C. dentata and bacteria.

Figure 4 Variation in spectroscopic indices (HIX, BIX, β/α, and FI) was observed among seven groups.

Each value represents the mean ± SD (n = 3). Different symbols (e.g., ** and ***) represented significant differences between the co-culture group of C. dentata and bacteria (p < 0.05, p < 0.001), respectively.

Principal component analysis

PCA was performed to visualize the characteristics and origins of DOM in the algae environment (Fig. 5). The PCA analysis included HIX, FI, BIX, β/α, C. dentata abundance, bacterial (CA6 and CF2) abundance, CDOM (C1 and C2) components, and Fv/Fm. The first and second principal components of PCA from the C. dentata-bacteria co-cultured (CA6 and CF2) group (Figs. 5A and 5D) accounted for 38.6% and 31.8%, 56.9% and 20.8%, respectively; for bacteria cultured by the filtrate group (CA6 and CF2) (Figs. 5B and 5E), they accounted for 56.8% and 30.5%, 61.3% and 19.4%, respectively; while for the axenic bacteria group (CA6 and CF2) (Figs. 5C and 5F), they accounted for 63.5% and 23.2%, 53.2% and 21.4%, respectively. Along PC2, Fv/Fm or CA6 loadings were clustered positively in Fig. 5A. The loadings of HIX, C2, and CA6 were clustered positively along PC1 in Fig. 5B. In Figs. 5C and 5E, the loadings of HIX and C2 were clustered positively along PC1. Meanwhile, CF2 and C1 were clustered and showed positive loadings along PC2 in Fig. 5D. In Fig. 5F, the loadings of HIX and C2 were clustered negatively along PC1.

Figure 5 The PCA loadings for the dataset of seven groups.

C. dentata co-cultured with (CA6/CF2) bacteria group (A, D); and filter cultured with (CA6/CF2) bacteria group (B, E); and axenic (CA6/CF2) bacterial group (C, F). The DOM indices parameters (FI, HIX, BIX, the PARACFAC components and β/α) and C. dentata abundance are indicated as vector arrows.

Discussion

Growth of microalgae and bacteria, chlorophyll fluorescence variables of C. dentata

Microalgae and microbial communities of their phycosphere constituted complex and dynamic ecosystems (Holmström et al., 2002). Many bacterial communities are beneficial for algal host (Cole, 1982). In the present study, the microalgae cells abundance of C. dentata-CA6 co-cultured group was significantly higher than the axenic C. dentata group at early stage of incubation. Meanwhile, the abundance of C. dentata-CF2 co-cultured group reached its maximum in the later stage of the culture. This suggests that C. dentata abundance is stimulated by CA6 and CF2 growth, and CA6 have a stronger ability to improve the C. dentata cell abundance than CF2. Recent research has found that the mode of marine bacteria life influences the utilization of DOM (Luo & Moran, 2015). On the other hand, PA bacteria exhibit higher enzymatic activity and cell-specific activity compared to FL bacteria (Murrell et al., 1999; Mével et al., 2008), and are capable of colonizing most phytoplankton or granular organic matter (Rieck et al., 2015). Meanwhile, some researchers found Roseobacter can promote algal growth (Seyedsayamdost et al., 2011), through its cells attached to the surface of the dinoflagellates Pfiesteria (Alavi et al., 2001) and Prorocentrumlima (Ehrenberg) F. Stein (Wagner-Döbler et al., 2010).

On the other hand, the Fv/Fm values are widely used to describe the photosynthetic properties of microalgae (Dabrowski et al., 2021). Photosynthetic parameters provide a comprehensive understanding of the effects of CA6 and CF2 on the performance of the C. dentata cell’s photosynthetic system. Numerous studies have shown that environmental stress leads to a decline in algal photosynthetic efficiency (Solovchenko et al., 2019). However, our experiments demonstrated that adding CA6 and CF2 to C. dentata cells increased their photosynthetic efficiency and prolonged their photosynthetic cycle. The photosynthetic indicators of microalgae reached their peak during mid-growth but decreased in later stages, indicating that the addition of CA6 or CF2 promoted the algae’s photosynthetic system. Meanwhile, the abundance of C. dentata cells reached its maximum in the late stage of culture, and the Fv/Fm value peaked in the middle stage (Figs. 1E and 1F). However, there was a weak correlation between C. dentata cell abundance and photosynthetic parameters in the C. dentata-bacteria co-cultured group (Figs. 5A and 5D). This suggests that bacteria have a certain delay in enhancing the microalgae cell abundance compared to their ability to enhance the photosynthetic properties of C. dentata. Meanwhile, previous research has shown that cell size is directly proportional to Fv/Fm (Patil & Anil, 2018), indicating that the C. dentata-bacteria co-cultured group in the middle stage exhibits larger cell sizes with smaller microalgal abundance, while in the late stage of culture, it shows smaller cell sizes with larger algal abundance.

Analysis of Chrysotila dentata-derived DOM

CA6 and CF2 were isolated from the C. dentata solution. CA6 belongs to the particle-attached (PA) bacteria of the C. dentata solution, while CF2 belongs to the free-living (FL) bacteria of the C. dentata solution. PCA analysis based on DOM indices, C. dentata abundance, and bacterial cell abundance showed that the first principal component (PC1) and the second principal component (PC2) explained 70.4–87.3% of the total variance (Fig. 5). Most of PC1 was positively correlated with most DOM parameters, such as Fmax of C2 and HIX, indicating that PC1 was associated with bacteria-transforming DOM processes in which the humic-like component increased while the protein-like component decreased (Figs. 5A–5E). Partial PC2 was positively correlated with the bacterial abundance of CF2 and CA6, suggesting that PC2 was associated with bacterial growth (Figs. 5C–5E). In this study, there were differences observed among experimental subgroups in terms of their ability to transform and utilize protein- and humic-like components by CA6 and CF2. However, in the C. dentata-bacteria co-cultured groups, the transformation and fluorescence properties of DOM were similar between the C. dentata solution treated with CF2 and CA6. In events B and E, PC1 positively correlated with HIX, C2, and C. dentata; in event B, PC2 positively correlated with the photosynthetic parameters of microalgae. In event E, PC2 was positively correlated with C1 and CF2. This indicated that DOM had a low concentration and high humification degree, which related to the cell abundance and high activity of C. dentata in the C. dentata-CA6 co-cultured group (Fig. 5B). Additionally, DOM had a high concentration and humification degree, which related to the cell abundance of C. dentata in the C. dentata-CF2 co-cultured group (Fig. 5E). Furthermore, our research found that the ability to transform humic-like substances was higher in the C. dentata-CA6 co-cultured group compared to the C. dentata-CF2 co-cultured group in incubation (Fig. 3C). Previous research has found that “unstable” DOM was generally transformed into recalcitrant DOM by bacterial metabolism (Kitayama, Hama & Yanagi, 2007). Liu et al. (2019b) found that PA bacteria were more sensitive to environmental changes than FL bacteria; PA bacteria have chemotaxis and motility and can couple with phytoplankton (Stocker & Seymour, 2012), rapidly utilizing organic matter patchiness (Luo & Moran, 2015) to maintain site-specific selection pressure, allowing bacteria to find and compete on densely populated surfaces (Luo & Moran, 2015). The results of this study confirm previous findings that CA6 has the ability to utilize and transform C. dentata-derived DOM more effectively than CF2.

Analysis of filtrate-derived DOM

In previous studies, it was found that bacteria preferentially consume “fresh” DOM produced by phytoplankton (Shimotori, Omori & Hama, 2009). Additionally, bacteria have the ability to convert unstable macromolecules into stable and highly aromatic humic substances of small size (Chen et al., 2021). We found that the transformation and fluorescence properties of DOM were also similar between the filtrate of C. dentata treated with CF2 and CA6. PC1 was positively correlated with CA6, C2, and HIX, and PC2 was positively correlated with BIX in event C (Fig. 5C). In event F, PC1 was positively correlated with C2 and HIX, while PC2 was positively correlated with CF2 and BIX (Fig. 5F). This suggests that CA6 and CF2 bacteria cultured with a filtrate group have a high capacity to transform protein-like components into humic substances. Furthermore, it was observed that CA6 and CF2 exhibited weaker abilities in consuming protein-like substances when cultured with filtrate. However, CA6 showed a stronger ability to convert C. dentata-derived humic-like substances (Fig. 3D). Liu, Wang & Sun (2022) also discovered that epiphytic bacteria exhibited a limited capacity to transform DOM derived from S. dohrnii filtrate. This indicates that DOM derived from C. dentata filtrate was not considered “fresh” DOM, but PA bacteria were able to convert C. dentata-derived DOM into humic-like components as well.

Analysis of bacteria-derived DOM

CF2 has a higher capacity to consume bacterial-derived protein-like components than CA6, but its ability to convert humic substances is weaker compared to CA6. In event D, the HIX, C2, β/α, clustered in the positive semiaxis of PC1, and PC2 were positively correlated with CA6 and negatively correlated with C1 (Fig. 5D). This indicates that DOM had a high concentration and high humification, and CA6 effectively utilized bacterial-derived protein-like components in the axenic CA6 cultured group. In event G, PC1 was positively correlated with BIX and negatively correlated with C2, HIX, and FI, while PC2 was positively correlated with β/α (Fig. 5G). These findings suggest that CF2 utilized bacterial-derived humic-like components.

Previous researchers have conducted corresponding experiments on the degradation of bacterial-derived DOM, in which bacteria were able to convert 10–15% of bacterial-derived DOM into humic substances lacking amino acid and carbohydrate characteristics (Ogawa et al., 2001). Bussmann (1999) discovered that humic-like components can also serve as a carbon source and provide energy to bacteria. Interestingly, in the present study, CF2 efficiently consumed bacterial-derived humic-like components to fuel its growth and metabolism by providing energy. This study demonstrated that CF2 exhibited a greater ability to utilize bacterial-derived DOM compared to CA6. Additionally, it suggests that phycosphere bacteria surrounding C. dentata display varying preferences for different sources of DOM. Particle-attached bacteria (CA6) showed a preference for C. dentata-derived DOM, whereas free-living bacteria (CF2) favored bacterial-derived DOM.

Conclusions

This study aimed to assess the physiological characteristics and soluble organic matter characteristics of C. dentata when co-cultured with Marinobacter hydrocarbonoclasticu as well as Bacillus firmus. When co-cultured with C. dentata, Marinobacter hydrocarbonoclasticus demonstrated a greater ability to promote the cell abundance and photosynthetic capacity of C. dentata compared to Bacillus firmus. Additionally, CDOM samples from the experimental group were tested, revealing two main fractions: protein-like and humic-like components during the 22-day incubation period. The transformation ability of different DOM components by Marinobacter hydrocarbonoclasticus and Bacillus firmus varied across different incubation groups. Specifically, Marinobacter hydrocarbonoclasticus exhibited a more significant capability in transforming C. dentata-derived protein-like components into humic-like substances compared to Bacillus firmus. Meanwhile, Marinobacter hydrocarbonoclasticus exhibited the strongest ability to convert into humic-like substances compared to other incubation groups when cultured in the filtrate of C. dentata. On the other hand, Bacillus firmus differs from humic-like components derived from C. dentata in that it was found to consume bacterial-derived humic-like substances. Additionally, there were certain limitations in selecting bacterial types and concentrations for this study. However, these findings provided supporting data on the interactions between C. dentata and Marinobacter hydrocarbonoclasticus as well as Bacillus firmus.

Supplemental Information

Supplemental Information 1 Raw data of EEM of Group 1 in 5.13

Click here for additional data file.

Supplemental Information 2 Raw data of EEM of Group 2 in 5.13

Click here for additional data file.

Supplemental Information 3 Raw data of EEM of Group 3 in 5.13

Click here for additional data file.

Supplemental Information 4 Raw data of EEM of Group 4 in 5.13

Click here for additional data file.

Supplemental Information 5 Raw data of EEM of Group 5 in 5.13

Click here for additional data file.

Supplemental Information 6 Raw data of EEM of Group 6 in 5.13

Click here for additional data file.

Supplemental Information 7 Raw data of EEM of Group 7 in 5.13

Click here for additional data file.

Supplemental Information 8 Raw data of EEM of Group 8 in 5.13

Click here for additional data file.

Supplemental Information 9 Raw data of EEM of Group 9 in 5.13

Click here for additional data file.

Supplemental Information 10 Raw data of EEM of Group 10 in 5.13

Click here for additional data file.

Supplemental Information 11 Raw data of EEM of Group 11 in 5.13

Click here for additional data file.

Supplemental Information 12 Raw data of EEM of Group 12 in 5.13

Click here for additional data file.

Supplemental Information 13 Raw data of EEM of Group 13 in 5.13

Click here for additional data file.

Supplemental Information 14 Raw data of EEM of Group 14 in 5.13

Click here for additional data file.

Additional Information and Declarations

Competing Interests

Author Contributions

Data Availability

The authors declare there are no competing interests.

xueru wang performed the experiments, analyzed the data, prepared figures and/or tables, authored or reviewed drafts of the article, and approved the final draft.

chenjuan fan performed the experiments, authored or reviewed drafts of the article, and approved the final draft.

Jun Sun conceived and designed the experiments, prepared figures and/or tables, and approved the final draft.

The following information was supplied regarding data availability:

The raw data are available in the Supplemental Files.

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
