# Peer review of "Utilization and transformation of Chrysotila dentata-derived dissolved organic matter by phycosphere bacteria Marinobacter hydrocarbonoclasticus and Bacillus firmus"

_PeerJ, doi:10.7717/peerj.16552_

## Round 0.1 · original submission · Major Revisions

The reviewers have noted a number of issues that should be addressed in a revised version of the text. In my opinion, the reviewer comments about English are a bit harsh, but I agree that the manuscript can be greatly improved in terms of clarifying for the reader how the experimental results directly lead to the stated conclusions and how they compare to the published literature. Revising the text so that it is more understandable to someone who is not already familiar with this project would be more important than consulting an English grammar expert. Both reviewers agree that the experimental work is of high quality, so a thorough revision of the text could make it worthy of publication.

**Language Note:** The review process has identified that the English language must be improved. PeerJ can provide language editing services - please contact us at [email protected] for pricing (be sure to provide your manuscript number and title). Alternatively, you should make your own arrangements to improve the language quality and provide details in your response letter. – PeerJ Staff

Reviewer 1 ·

Basic reporting

I have some comment for your manuscript.
1. Introduction:
Why do you chose Marinobacter hydrocarbonclasticus and Bacillus firms to investigate?
"To explore the transformation of C.dentata-derived DOM by significant bacteria in the Bohai Sea water of China, we examined..." - I didn't see the relationship between the Bohai Sea water of China and your research. Do you mean you collect samples from Bohai Sea water to perform the experiment or just isolated microorganisms?
2. Materials and Methods:
Line 110: Marinobacter hydrocarboclaticus and Bacillus firms were isolated from the C.dentata solution...Is it correct? or were they isolated from sea water?
3. Results
Line 187: "This day was marked as 0 days". Could you explain this sentence? you mean the day you added CA6 and CF2 marked as 0? so the next sentence: "After 22 days of cultivation, the C. dentata group declined from..." - it means 22 days after you add CA6 and CF2?
4. Discussion:
Could you explain the trends of figures 1? Why the C.dentata and CA6 bacteria co-culture group increased after 22 days of cultivation, while the co-culture group of C.dentata and bacteria CF2 decreased? Are there any nutrient competition, references for your discussion, ect?

Experimental design

No comment

Validity of the findings

No comment

Additional comments

Carefully check the spelling. and grammar. For examples, line 258, 260 "The" instead of "the"

Reviewer 2 ·

Basic reporting

The manuscript “Utilization and transformation of Chrysotila dentata-derived dissolved organic matter by phycosphere bacteria Marinobacter hydrocarbonoclasticus and Bacillus firmus” by Xueru Wang et al. describes the utilization of microalgae and bacterial derived DOM by the two mentioned heterotrophic bacteria. Despite of potential interesting results, the form of the manuscript (language, structure of paragraphs, storyline…) does not allow its publication.

Experimental design

Due to the language and (missing) structure of the article I could not evaluate the experimental design.

Validity of the findings

Please see above.

Additional comments

Affiliations: Remove the second “guangzhou, China”
Abstract (no line numbers)
- Please define CA6 and CF2. Are these the strains?
- Which influence? Of the DOM on the two bacteria (e.g. the growth, bacterial production, alkaline phosphatase…)?
- If you define CA6 and CF2 before you don’t need to name the exact strain again.
- “The protein-like component of C. dentata cultures added CA6 and CF2 bacteria were eûectively utilized.” Please rewrite. -> …were effectively utilized by CA6 and CF2?
- “The humic-like components were increased in cultures time.” -> just increased.
- What other than the filtrate group exist?
- “Finally, the ability of CA6 to transform microalgal-derived DOM was superior to that of a CF2, and CF2 to consume bacterial derived DOM was superior to that of CA6.” -> based on which results, based on which method?

Introduction:
Please rewrite the introduction with a skilled English-speaking person. It is generally hard to follow and does not guide the reader towards your main topics.
Start with the importance of DOM. Next: State that DOM is composed of different molecules. Name the reasons for the differential composition. Guide to your questions.
Few examples:
L46-47: Please rewrite. Maybe: Marine dissolved organic carbon (DOM) is one of the largest reservoirs of organic carbon in the global carbon cycle, and consists of a mixture of molecules of different molecular weights (reference).
L48-49: Pleas rewrite. It is hard to follow.
L54: What is marine fluorescence?
L65: Does not every photosynthetic eukaryote produce organic carbon?
L67: What are living environments?
L68-69: Please write full sentences.
L69-73: Please write a text with coherent sentences that built up a story and guide the reader, and not only string words together .
L74-75: Please rewrite. One can only guess what you want to say.

Materials and Methods:
I cannot understand what you did in the experiments, which makes a fair evaluation of your work impossible. Please cooperate with a person that is skilled in the English language.
L159: What is this symbol?

L379-380: “Attached bacteria (CA6) preferred microalgal-derived DOM, while free-living bacteria favored bacterial-derived DOM”. Nice result! But I cannot find it in any Fig.nor the experimental procedure that should be the basis for this.

Figure 1: The different symbols are hard to differentiate! What are panels D, E and F? They are not mentioned in the fig caption.

Full article:
Please cooperate with a person that is skilled in the English language, and try to write a coherent text and make it a story. I think you have some interesting results (especially the uptake of humic-like bacterial derived DOM by other bacteria), but it is really hard to follow.

---

## Round 0.2 · accepted · Accept

The primary concern during the review was that the text was not clear enough to understand the methods and results. The authors have extensively rewritten and edited the manuscript for clarity, and it is much improved. The reviewers have declined to review the manuscript again, so I am accepting the manuscript for publication.